# SIMPLE BLACK-BOX ADVERSARIAL ATTACKS

## ABSTRACT

The construction of adversarial images is a search problem in high dimensions within a small region around a target image. The goal is to find an imperceptibly modified image that is misclassified by a target model. In the black-box setting, only sporadic feedback is provided through occasional model evaluations. In this paper we provide a new algorithm whose search strategy is based on an intriguingly simple iterative principle: We randomly pick a low frequency component of the discrete cosine transform (DCT) and either add or subtract it to the target image. Model evaluations are only required to identify whether an operation decreases the adversarial loss. Despite its simplicity, the proposed method can be used for targeted and untargeted attacks — resulting in previously unprecedented query efficiency in both settings. We require a median of 600 black-box model queries (ResNet-50) to produce an adversarial ImageNet image, and we successfully attack Google Cloud Vision with 2500 median queries, averaging to a cost of only \$3 per image. We argue that our proposed algorithm should serve as a strong baseline for future adversarial black-box attacks, in particular because it is extremely fast and can be implemented in less than 20 lines of PyTorch code.

## 1 INTRODUCTION

As machine learning systems become prevalent in numerous application domains, the security of these systems in the presence of malicious adversaries becomes an important area of research. Many recent studies have shown that decisions output by machine learning models can be altered arbitrarily with imperceptible changes to the input (Carlini & Wagner, 2017b). These attacks on machine learning models can be categorized by the capabilities of the adversary. *White-box* attacks require the adversary to have complete knowledge of the target model, whereas *black-box* attacks require only queries to the target model that may return complete or partial information.

Although often misunderstood as a property of neural networks (Szegedy et al., 2014), the vulnerability towards adversarial examples is likely an inevitability of classifiers in high-dimensional spaces (Shafahi et al., 2018). When the space is high-dimensional, randomly sampled points tend to be far apart (simply because to be close, they would have to be similar along every dimension). However, the reverse is true for distances between points and high-dimensional decision surfaces. The distance from a point to a hyper-plane is only measured along the direction of its normal vector (all other orthogonal directions are parallel to the hyper-plane).[1] One can of course construct pathological counter examples of distributions with two classes that have only non-zero support in small but distant regions of the space (e.g. classifying all-white vs. all-black images). However, the fact that almost all models for classification of natural images are susceptible to white-box attacks (Athalye et al., 2018), suggests that natural images tend to be very close to decision boundaries learned by classifiers and demonstrates that the distribution of natural images is no such exception.

If adversarial examples (almost) always exist, attacking a classifier turns into a search problem within a small volume around a target image. In the white-box scenario, this search can be guided effectively with gradient descent (Szegedy et al., 2014; Carlini & Wagner, 2017b; Madry et al., 2017). However, the black-box threat model is more applicable in many scenarios. Most popular online machine learning services such as Clarifai or Google Cloud Vision only allow API calls to access the model's predictions. These services do not release any internal details such as training data and model parameters. Moreover, queries to the model may incur a significant cost of both time

---

[1]Similar results hold for non-linear decision surfaces (Shafahi et al., 2018).

and money. The number of black-box queries made to the model therefore serves as an important metric of efficiency for the attack algorithm. Attacks that are too costly, or are easily defeated by query limiting, pose less of a security risk than efficient attacks. To date, the average number of queries performed by the best known black-box attacks remains high despite a large amount of recent work in this area (Chen et al., 2017; Brendel et al., 2017; Cheng et al., 2018; Guo et al., 2018; Tu et al., 2018; Ilyas et al., 2018). The most efficient and complex attacks still typically require tens or even hundreds of thousands of queries. Until now, a method for query efficient black-box attacks has remained an open research problem.

In this paper we propose a simple, yet highly efficient black-box attack that is based on a very simple intuition: If the distance to a decision boundary is very small, we don't have to be too careful about the exact direction along which we traverse towards it. Concretely, we repeatedly pick a *random* direction among a pre-specified set of orthogonal search directions, check if it is pointing towards or away from the decision boundary, and perturb the image by adding or subtracting the vector from the image. Each update moves the image further away from the original image and towards the decision boundary.

We provide some theoretical insight on the efficacy of our approach and evaluate various orthogonal search subspaces. Similar to Guo et al. (2018), we observe that restricting the search towards the low frequency end of the discrete cosine transform (DCT) basis is particularly query efficient. Further, we demonstrate empirically that our approach achieves a similar success rate to state-of-the-art black-box attack algorithms, however with an unprecedented low number of black-box queries. Due to its simplicity — it can be implemented in PyTorch in under 20 lines of code — we consider our method a new and perhaps surprisingly strong baseline for adversarial image attacks and we refer to our algorithm as *Simple Black-box Attack (SimBA)*.

## 2  BACKGROUND

The study of adversarial examples concerns with the robustness of a machine learning model to small changes in the input. In the realm of classification, given a model $h$ and some input-label pair $(\mathbf{x}, y)$ on which the model correctly classifies $h(\mathbf{x}) = y$, $h$ is said to be $\rho$-robust with respect to perceptibility metric $d(\cdot, \cdot)$ if $h(\mathbf{x}') = y$ for all $\mathbf{x}'$ such that $d(\mathbf{x}, \mathbf{x}') < \rho$. The metric $d$ is often approximated by the $L_0$, $L_2$ and $L_\infty$ distances to measure the degree of visual dissimilarity between the clean input $\mathbf{x}$ and the perturbed input $\mathbf{x}'$. Following (Moosavi-Dezfooli et al., 2016; Moosavi-Dezfooli et al., 2017), for the remainder of this paper, we will use $d(\mathbf{x}, \mathbf{x}') = \|\mathbf{x} - \mathbf{x}'\|_2$ as the perceptibility metric unless specified otherwise.

Although this robustness requirement often holds when $\mathbf{x}' = \mathbf{x} + \delta$ for randomly sampled noise vectors $\delta$, many studies have shown that the model admits directions of non-robustness even for very small values of $\rho$ (Moosavi-Dezfooli et al., 2016; Carlini & Wagner, 2017b). More recent work (Shafahi et al., 2018) has verified this claim theoretically, showing that adversarial examples are inherent in high-dimensional spaces. These findings motivate the problem of finding adversarial directions $\delta$ that alter the model's decision.

The simplest success condition for the adversary is to change the original correct prediction of the model to an arbitrary class, i.e., $h(\mathbf{x}') \neq y$. This is known as an *untargeted attack*. In contrast, a *targeted attack* aims to construct $\mathbf{x}'$ such that $h(\mathbf{x}') = y'$ for some chosen target class $y'$. For the sake of brevity, we will focus on untargeted attacks in our discussion, but all arguments in our paper are applicable to targeted attacks as well.

Since the model outputs discrete decisions, finding adversarial perturbations to change the model's prediction is, at first, a discrete optimization problem. However, it is often useful to define a surrogate loss $\ell_y(\cdot)$ that measures the degree of certainty that the model $h$ classifies the input as class $y$. The adversarial perturbation problem can therefore be formulated as the following constrained (continuous) optimization problem:

$$\min_{\delta} \ \ell_y(\mathbf{x} + \delta)$$

$$\text{subject to } \|\delta\|_2 < \rho.$$

When the model $h$ also outputs probabilities $p_h(\cdot \mid \mathbf{x})$ associated with each class, one commonly used adversarial loss is the probability of class $y$: $\ell_y(\mathbf{x}') = p_h(y \mid \mathbf{x}')$. For targeted attacks towards label $y'$ a common choice is $\ell_{y'}(\mathbf{x}') = -p_h(y' \mid \mathbf{x}')$

Depending on the application domain, the attacker may have various degrees of knowledge about the target model $h$. Under the *white-box* threat model, the classifier $h$ is provided to the adversary. In this scenario, a powerful attack strategy is to perform gradient descent on the adversarial loss $\ell_y(\cdot)$, or an approximation thereof. The perturbation norm $\|\delta\|_2$ can be controlled by early stopping (Goodfellow et al., 2015; Kurakin et al., 2016) or by including it as a component of the loss function (Carlini & Wagner, 2017b).

However, the white-box assumption may be unsuitable for many applications. For instance, the model $h$ may be exposed to the public as an API, allowing only queries on inputs. This *black-box* threat model is much more challenging for the adversary, since gradient information may not be used to guide the finding of the adversarial direction $\delta$, and each query to the model incurs a time and monetary cost. Thus, the adversary is tasked with an additional goal of minimizing the number of black-box queries to $h$ while succeeding in constructing an imperceptible adversarial perturbation. This poses a slightly modified constrained optimization problem (with a slight abuse of notation):

$$\min_{\delta} \ \ell_y(\mathbf{x} + \delta)$$
$$\text{subject to} \ \|\delta\|_2 < \rho \tag{1}$$
$$\text{queries} \leq B$$

where $B$ is some fixed budget for the number of queries allowed during the optimization. For iterative methods, the budget $B$ constrains the number of iterations the algorithm may take, hence requiring that the attack algorithm converges to a solution very quickly.

## 3 A SIMPLE BLACK-BOX ATTACK

Imagine that we have some image $\mathbf{x}$ which a black-box neural network classifies as category $y$ with predicted probability $p_h(y \mid \mathbf{x})$. Our goal is to find a small pertubation $\delta$ such that the prediction $h(\mathbf{x} + \delta) \neq y$.

---

**Algorithm 1** SimBA in Pesudocode

1: **procedure** SIMBA($\mathbf{x}, y, Q, \epsilon$)
2:    $\delta = \mathbf{0}$
3:    $\mathbf{p} = p_h(y \mid \mathbf{x})$               ▷ Query black-box for initial probabilities
4:    **while** $\mathbf{p}_y = \max_{y'} \mathbf{p}_{y'}$ **do**          ▷ While the true label has not changed
5:        Pick randomly without replacement: $\mathbf{q} \in Q$
6:        **for** $\alpha \in \{\epsilon, -\epsilon\}$ **do**         ▷ Try $\epsilon\mathbf{q}$ and then $-\epsilon\mathbf{q}$ if that fails.
7:            $\mathbf{p}' = p_h(y \mid \mathbf{x} + \delta + \alpha\mathbf{q})$         ▷ Query the black-box model
8:            **if** $\mathbf{p}'_y < \mathbf{p}_y$ **then**        ▷ If the probability is lower, accept this step
9:                $\delta = \delta + \alpha\mathbf{q}$
10:               $\mathbf{p} = \mathbf{p}'$
11:               **break**                ▷ Move on to the next direction
         **return** $\delta$

---

The intuition behind our method is simple: for *any* direction $\mathbf{q}$ and some step size $\epsilon$, one of $\mathbf{x} + \epsilon\mathbf{q}$ or $\mathbf{x} - \epsilon\mathbf{q}$ is likely to decrease $p_h(y \mid \mathbf{x})$. We therefore repeatedly pick random directions $\mathbf{q}$ and either add or subtract them. The approach is summarized in psuedocode in Algorithm 1. To minimize the number of queries to $h(\cdot)$ we always first try adding $\epsilon\mathbf{q}$. If this decreases the probability $p_h(y \mid \mathbf{x})$ we take the step, otherwise we try subtracting $\epsilon\mathbf{q}$. This procedure requires between 1.4 and 1.5 queries per update on average (depending on the data set and target model). Our proposed method – *Simple Black-box Attack* (SimBA) – takes as input the target image label pair $(\mathbf{x}, y)$, a set of orthogonal candidate vectors $Q$ and a step-size $\epsilon > 0$. For simplicity we pick $\mathbf{q} \in Q$ uniformly at random. To guarantee maximum query efficiency, we ensure that no two directions cancel each other out and diminish progress, or amplify each other and increase the norm of $\delta$ disproportionately. For this reason we pick $\mathbf{q}$ *without replacement* and restrict all vectors in $Q$ to be *orthogonal*. As we show later, this results in a guaranteed perturbation norm of $\|\delta\|_2 = \sqrt{T}\epsilon$ after $T$ updates.

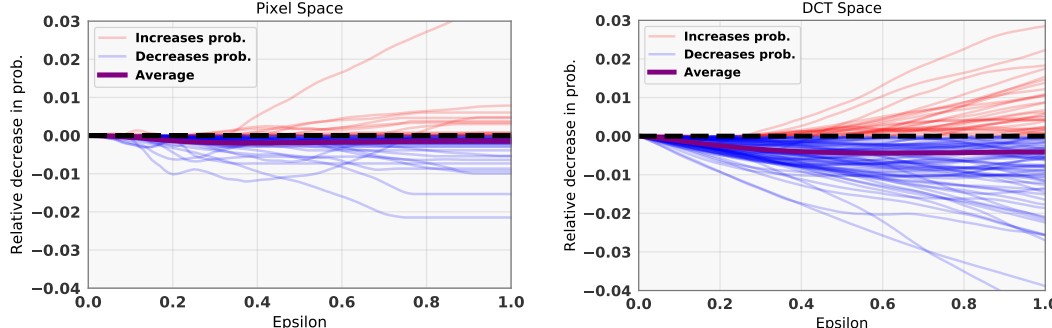

Figure 1: Plot of the change in true class probability when descending in a random direction in pixel space (left) and DCT space (right) at step size $\epsilon$. The average change (purple line) is almost linear in $\epsilon$ with the slope being steeper when the direction is sampled in DCT space. Furthermore, 98% of the directions sampled in DCT space have either $\mathbf{q}$ or $-\mathbf{q}$ descending, while only 73% are descending in pixel space.

### 3.1 HYPER-PARAMETERS

The only hyper-parameters of SimBA are the set of orthogonal search vectors $Q$ and the step size $\epsilon$.

**Orthogonal directions $Q$.** A natural first choice for the set of orthogonal search directions $Q$ is to randomly subsample the natural basis, $Q = I$, which corresponds to performing our algorithm directly in pixel space. Essentially each iteration we are increasing or decreasing one color of a single randomly chosen pixel. Alternatively, we can consider directions in frequency domain. Recent work has discovered that random noise in low frequency space are more likely to be adversarial (Guo et al., 2018). To exploit this fact, we follow Guo et al. (2018) and propose to exploit the *discrete cosine transform* (DCT). The discrete cosine transform is an orthonormal transformation that maps signals in a 2D image space $\mathbb{R}^{d \times d}$ to frequency coefficients corresponding to magnitudes of wave functions. In what follows, we will refer to the set of orthonormal frequencies extracted by the DCT as $Q_{\mathrm{DCT}}$. While the full set of directions $Q_{\mathrm{DCT}}$ contains $d \times d$ frequencies, we keep only a fraction $r$ of the lowest frequency directions in order to make the adversarial perturbation in the low frequency space.

**Choosing a learning rate $\epsilon$.** Given any set of search directions $Q$, some directions may decrease $p_h(y \mid \mathbf{x})$ more than others. Furthermore, it is not naively obvious that arbitrary steps along any direction $\mathbf{q}_i$ decrease the probability. It is possible for the probability $p_h(y \mid \mathbf{x} + \epsilon \mathbf{q}_i)$ to be non-monotonic in $\epsilon$. In Figure 1, we plot the relative decrease in probability as a function of $\epsilon$ for randomly sampled search directions in both pixel space and the DCT space. This figure highlights an illuminating result: the probability $p_h(y \mid \mathbf{x} \pm \epsilon \mathbf{q})$ *decreases monotonically* in $\epsilon$ with surprising consistency (across random images and vectors $\mathbf{q}$)! Although some directions eventually increase the true class probability, the expected change in this probability is negative with a relatively steep slope. This means that our algorithm is not overly sensitive to the choice of $\epsilon$ and the iterates will decrease the true class probability quickly. The figure also shows that search in the DCT space tends to lead to steeper descent directions than pixel space. As we show in the next section, we can tightly bound the final $L_2$-norm of the perturbation given a choice of $\epsilon$ and maximum number of steps $T$, so the choice of $\epsilon$ depends *primarily on budget considerations* with respect to $\|\delta\|_2$.

### 3.2 BUDGET CONSIDERATIONS.

Each iteration of our algorithm strictly increases the norm of the adversarial perturbation $\delta_i = \epsilon \mathbf{q}_i$. By exploiting the orthonormality of the basis $Q$, we can bound this change tightly. Let $\alpha_i \in \{-\epsilon, 0, \epsilon\}$ denote the sign of the search direction chosen at step $i$, so

$$\delta_{i+1} = \delta_i + \alpha_i \mathbf{q}_i.$$

We can recursively expand $\delta_{i+1} = \delta_{i-1} + \alpha_{i-1}\mathbf{q}_{i-1} + \alpha_i\mathbf{q}_i$. In general, the final perturbation $\delta_T$ after $T$ steps can be written as a sum of these individual search directions:

$$\delta_T = \sum_{i=1}^{T} \alpha_i \mathbf{q}_i.$$

Since the directions $\mathbf{q}_i$ are orthogonal, $\mathbf{q}_i^\top \mathbf{q}_j = 0$ for any $i \neq j$. We can therefore compute the $L_2$-norm of the adversarial perturbation:

$$\|\delta_T\|_2^2 = \left\|\sum_{i=1}^{T} \alpha_i \mathbf{q}_i\right\|_2^2 = \sum_{i=1}^{T} \|\alpha_i \mathbf{q}_i\|_2^2 = \sum_{i=1}^{T} \alpha_i^2 \|\mathbf{q}_i\|_2^2 \leq T\epsilon^2.$$

Here, the second equality follows from the orthogonality of $\mathbf{q}_i$ and $\mathbf{q}_j$, and the last inequality is tight if all queries result in a step of either $\epsilon$ or $-\epsilon$. Thus the adversarial perturbation has $L_2$-norm at most $\sqrt{T}\epsilon$ after $T$ iterations. The same analysis holds when using any orthonormal basis (e.g. $Q_{\mathrm{DCT}}$).

This result highlights an important trade-off for our method. For query-limited scenarios, we may reduce the number of iterations by setting $\epsilon$ higher, incurring higher perturbation $L_2$-norm. If a low norm solution is more desirable, reducing $\epsilon$ will allow quadratically more queries at the same $L_2$-norm. A more thorough theoretical analysis of this trade-off could improve query efficiency.

## 4 EXPERIMENTAL EVALUATION

In this section, we empirically evaluate our attack against a comprehensive list of known black-box attack algorithms: ZOO (Chen et al., 2017), the Boundary Attack (Brendel et al., 2017), Opt attack (Cheng et al., 2018), Low Frequency Boundary Attack (LFBA) (Guo et al., 2018), AutoZOOM (Tu et al., 2018), and the QL attack (Ilyas et al., 2018). Based on Eq. (**??**), there are three dimensions to evaluate black-box adversarial attacks on: how often the optimization problem finds a feasible point (*success rate*), how many queries were required ($B$), and the resulting perturbation norms ($\rho$).

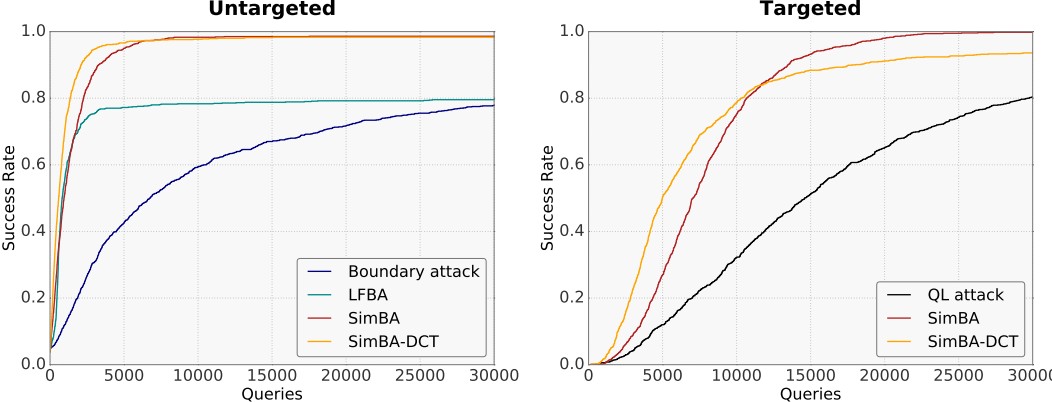

Figure 2: Comparison of success rate versus number of model queries for untargeted (left) and targeted (right) attacks. Horizontal axis shows queries in log scale. For boundary attack and LFBA, success is defined as achieving a perturbation $L_2$-norm of 10 or less. Note that both SimBA and SimBA-DCT achieve a high success rate very quickly.

### 4.1 SETUP

We first compare all methods on the ImageNet validation set. We present numbers reported by the original authors' papers in comparison to ours where applicable. For methods where the required values were unavailable (the boundary attack, LFBA, and the QL attack), we evaluate these methods ourselves using hyperparameters suggested by the authors. For all original and reproduced results, we sample a set of 1000 images from the ImageNet validation set that are initially classified correctly to avoid artificially inflating the success rate.

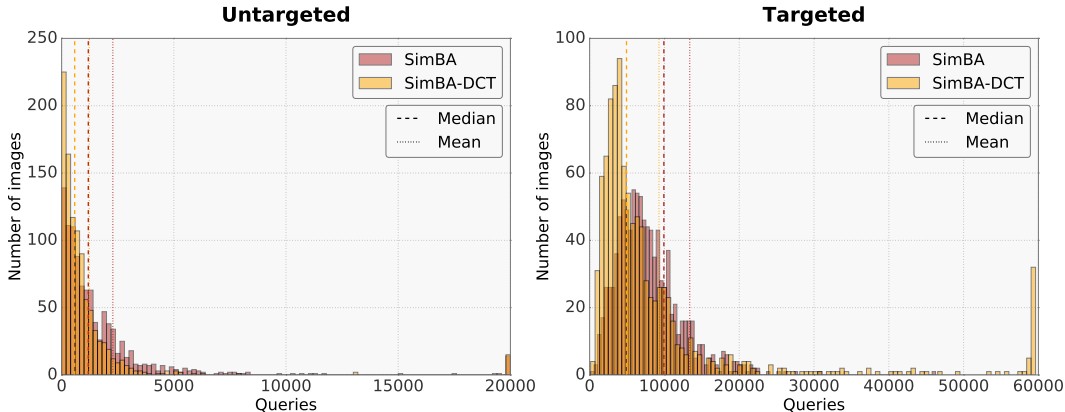

Figure 3: Histogram of queries required until a successful model attack (over 1000 target images). SimBA-DCT is highly skew-right, suggesting that only a handful of images require more than a small number of queries. The *median* number of queries required by SimBA-DCT is only 582.

| | Untargeted | | |
|---|---|---|---|
| **Attack** | **Average queries** | **Success rate** | **Average $L_2$** |
| ZOO | 192,000 | 88.9% | 1.20 |
| Boundary attack | 123,407 | 100% | 5.98 |
| Opt-attack | 71,100 | 100% | 6.98 |
| LFBA | 30,000 | 100% | 6.34 |
| SimBA | 1,665 | 98.6% | 3.77 |
| SimBA-DCT | **1,232** | 98.3% | 3.09 |

| | Targeted | | |
|---|---|---|---|
| **Attack** | **Average queries** | **Success rate** | **Average $L_2$** |
| QL-attack | 20,614 | 98.7% | 11.39 |
| AutoZOOM | 13,525 | 100% | 26.74 |
| SimBA | **7,899** | 100% | 8.83 |
| SimBA-DCT | 9,182 | 96.0% | 7.02 |

Table 1: Average query count for untargeted (left) and targeted (right) attacks. Both SimBA and SimBA-DCT have comparable success rate to other algorithms, but require a significantly fewer number of model queries. This effect is especially dramatic for untargeted attack, where the number of queries made by SimBA-DCT is 24 times fewer than all other methods.

We also evaluate SimBA in the real-world setting of attacking the Google Cloud Vision API. Due to the extreme budget required by baselines that might cost up to \$300 per image[2], we compare here only to LFBA, which we found to be the most query efficient baseline.

In our experiments, we limit SimBA and SimBA-DCT to at most $T = 10,000$ iterations for untargeted attacks and to $T = 30,000$ for targeted attacks. For SimBA-DCT, we use we keep the first 1/8th of all frequencies, and add an additional 1/32nd of the frequencies whenever we run out of frequencies without succeeding. For both methods, we use a fixed step size of $\epsilon = 0.2$.

### 4.2 IMAGENET RESULTS

**Evaluating success rate (Figure 2).** We first demonstrate the query efficiency of our method by showing that the average success rate increases dramatically faster than other methods with the same number of queries. Figure 2 compares SimBA and SimBA-DCT to the boundary attack and LFBA in the untargeted setting, and to the QL attack for the targeted setting. Since both boundary attack and LFBA are initialized with inputs of the adversarial class and gradually reduce $L_2$ perturbation norm, we define success as reducing the $L_2$-norm to 10 or below. For untargeted attack (left plot), the success rate for both SimBA and SimBA-DCT increase very quickly. LFBA is competitive with SimBA and SimBA-DCT until iteration 1000, but the final success rate is much worse than both methods. A similar result can be seen for targeted attack (right plot), where SimBA and SimBA-DCT outperform QL attack dramatically in terms of query efficiency.

**Query distributions.** In Figure 3 we plot the histogram of model queries made by both SimBA and SimBA-DCT over 1000 random images. Notice that the distributions are highly skewed so the median query count is much smaller than the average query count reported in Table 1. These median counts for SimBA and SimBA-DCT are only 944 and 582, respectively. In the targeted case,

---

[2]The Google API charges \$1.50 for 1000 image queries.

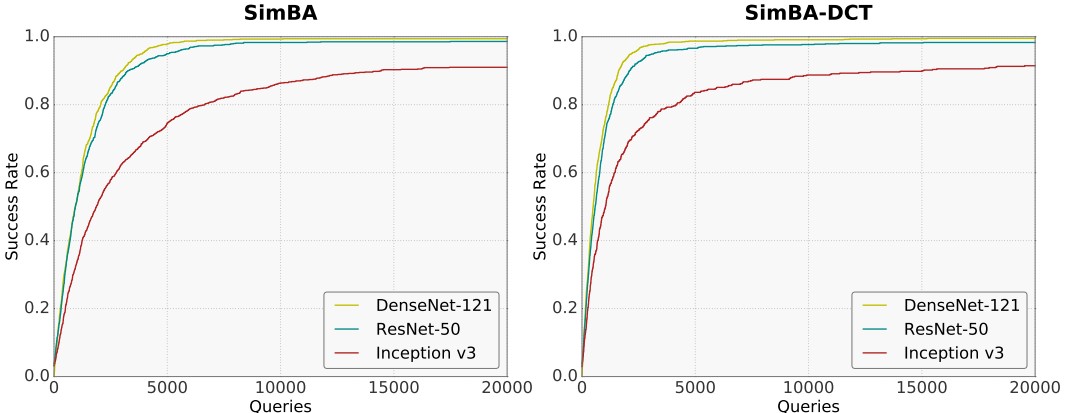

Figure 4: Comparison of success rate versus number of model queries across different network architectures for untargeted SimBA (left) and SimBA-DCT (right) attacks. For all networks, SimBA-DCT is more query efficient by at least a factor of 2. DenseNet is the most vulnerable against both attacks, admitting a success rate of almost $100\%$ after only 10,000 queries for SimBA and 4000 queries for SimBA-DCT. Inception v3 is much more difficult to attack for both methods.

while the majority of images can be successfully attacked with very few model evaluations, SimBA-DCT failed to find an adversarial perturbation for approximately $2.5\%$ of the images. Nevertheless, SimBA is achieves a success rate of $100\%$ within 7,899 average queries.

**Aggregate statistics (Table 1).** Table 1 computes aggregate statistics of model queries, success rate, and perturbation $L_2$-norm across different attack algorithms. The target model is a pretrained ResNet-50 (He et al., 2016) network, with the exception of AutoZOOM, which used an Inception v3 (Szegedy et al., 2016) network. All methods achieve a similar success rate except for ZOO, which has a reported success rate of $88.9\%$. The success rate for boundary attack and LFBA are always $100\%$ since both methods begin with very large perturbations to guarantee misclassification and gradually reduce the perturbation norm. Both SimBA and SimBA-DCT have lower average $L_2$-norm than all methods except for ZOO for untargeted attacks, while requiring an order of magnitude fewer queries (at 1665 and 1232, respectively). For targeted attack (right table), all the evaluated methods have much more similar query count, but both SimBA and SimBA-DCT still require significantly fewer queries than QL attack and Auto-ZOOM, and the $L_2$-norm of constructed perturbation is also much lower.

**Evaluating different networks (Figure 4).** To verify that our attack is robust against different model architectures, we evaluate SimBA and SimBA-DCT additionally against DenseNet-121 (Huang et al., 2017a) and Inception v3 (Szegedy et al., 2016) networks. Figure 4 shows success rate across the number of model queries for an untargeted attack against

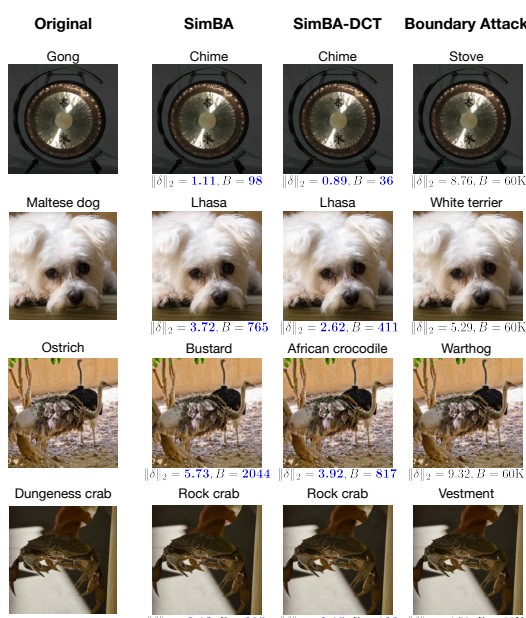

Figure 5: Randomly selected images before and after adversarial perturbation by SimBA, SimBA-DCT and boundary attack. The constructed perturbation is imperceptible for all three methods, but the $L_2$-norm for SimBA and SimBA-DCT is significantly lower than boundary attack across all images, despite allowing boundary attack to make 60,000 queries. In comparison, our methods are capable of constructing an adversarial example in as few as 36 queries. Zoom in for detail.

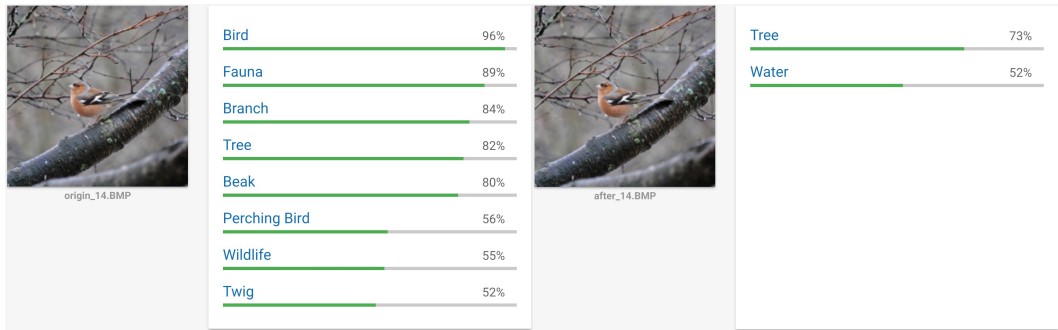

Figure 6: Screenshot of Google Cloud Vision labeling results on a randomly chosen image before and after adversarial perturbation. While the labels for the adversarial image are still reasonable, the concept *bird* has been completely removed. See Supplementary Material for additional samples.

the three different network architectures. ResNet-50 and DenseNet-121 exhibit a similar degree of vulnerability against our attacks. However, Inception v3 is noticeably more difficult to attack, requiring more than 10,000 queries to successfully attack with some images.

**Qualitative results (Figure 5).** For qualitative evaluation of our method, we present several randomly selected images before and after adversarial perturbation by untargeted attack. For comparison, we attack the same set of images using boundary attack for 30,000 iterations (60,000 model queries). Figure 5 shows the clean and perturbed images along with the perturbation $L_2$-norm and number of queries. While all attacks are highly successful at changing the label, SimBA and SimBA-DCT require much fewer queries and the resulting perturbation norm is also much smaller than that of boundary attack. In fact, SimBA-DCT was able to find an adversarial image in as few as 36 model queries! Notice that the perturbation produced by SimBA contains sparse but sharp differences, constituting a low $L_0$ norm but relatively high $L_\infty$ norm. However, SimBA-DCT produces perturbations that are sparse in frequency space, but the resulting change in pixel space is spread out across all pixels.

### 4.3 GOOGLE CLOUD VISION ATTACK

To demonstrate the effectiveness of our attack against real world systems, we attack the Google Cloud Vision API, an online machine learning service that provides labels for arbitrary input images. For a given image, the API returns a list of top concepts contained in the image and their associated probabilities. Since the full list of probabilities associated with every label is unavailable, we define an untargeted attack that aims to remove the top 3 concepts in the original. We use the maximum of the original top 3 concepts' returned probabilities as the adversarial loss and use SimBA to minimize this loss. Figure 6 shows a sample random image before and after the attack. The top 3 concepts, in particular the concept bird, have been removed in the adversarial image.

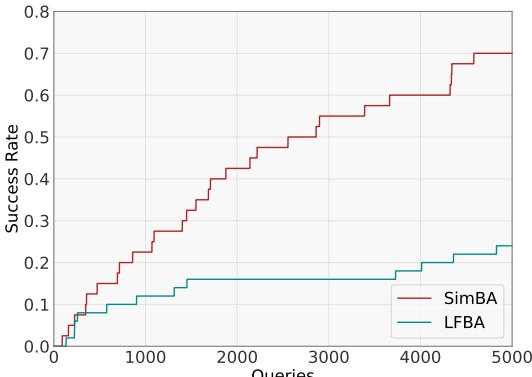

Figure 7: Plot of success rate across number of model queries for Google Cloud Vision attack. SimBA is able to achieve close to $70\%$ success rate after only 5000 queries, while the success rate for LFBA has only reached $25\%$.

Since our attack can be executed efficiently, we evaluate its effectiveness over an aggregate of 50 random images. For the LFBA baseline, we define an attack as successful if the produced perturbation has an $L_2$-norm of at most the highest $L_2$-norm in a successful run of our attack. Figure 7 shows the average success rate of both attacks across number of queries. SimBA achieves a final success rate of $70\%$ after only 5000 API calls, while LFBA is able to succeed only $25\%$ of the time under the same query budge. To the best of our

knowledge, this is the first adversarial attack result on Google Cloud Vision that has a high reported success rate within very limited number of queries.

## 5 RELATED WORK

Many recent studies have shown that both white-box and black-box attacks can be applied to a diverse set of tasks. Computer vision models for image segmentation and object detection have also been shown to be vulnerable against adversarial perturbations (Cisse et al., 2017a; Xie et al., 2017). Carlini & Wagner (2018) performed a systematic study of speech recognition attacks and showed that robust adversarial examples that alter the transcription model to output arbitrary target phrases can be constructed. Attacks on neural network policies (Huang et al., 2017b; Behzadan & Munir, 2017) have also been shown to be permissible.

As these attacks become prevalent, many recent works have focused on designing defenses against adversarial examples. One common class of defenses applies an image transformation prior to classification, which aims to remove the adversarial perturbation without changing the image content (Xu et al., 2017; Dziugaite et al., 2016; Guo et al., 2017). Instead of requiring the model to correctly classify all adversarial images, another strategy is to detect the attack and output an adversarial class when certain statistics of the input appear abnormal (Li & Li, 2017; Metzen et al., 2017; Meng & Chen, 2017; Lu et al., 2017). The training procedure can also be strengthened by including the adversarial loss as an implicit or explicit regularizer to promote robustness against adversarial perturbations (Tramèr et al., 2017; Madry et al., 2017; Cisse et al., 2017b). While these defenses have shown great success against a passive adversary, almost all of them can be easily defeated by modifying the attack strategy (Carlini & Wagner, 2017a; Athalye & Carlini, 2018; Athalye et al., 2018).

Relative to defenses against white-box attacks, few studies have focused on defending against adversaries that may only access the model via black-box queries. While transfer attacks can be effectively mitigated by methods such as ensemble adversarial training (Tramèr et al., 2017) and image transformation (Guo et al., 2017), it is unknown whether existing defense strategies can be applied to adaptive adversaries that may access the model via queries. Guo et al. (2018) have shown that the boundary attack is susceptible to image transformations that quantize the decision boundary, but employing the attack in low frequency space can successfully circumvent these transformation defenses. Given the query efficiency and real world applicability of our proposed black-box attacks, we hope that more research can be dedicated towards defending against malicious adversaries under this threat model.

## 6 CONCLUSION

We proposed SimBA, a simple black-box adversarial attack that takes small steps iteratively towards the decision boundary, and demonstrated through extensive experiment its unprecedented query efficiency in both the untargeted and targeted settings. Due to its accessibility, we hope that this method establishes a strong baseline for future research on black-box adversarial examples.

While we intentionally avoid more sophisticated techniques to improve the method in favor of simplicity, we believe that additional modifications can still dramatically decrease the number of model queries. One possible extension could be to further investigate the selection of different sets of orthonormal bases, which could be crucial to the efficiency of our method by increasing the probability of finding a direction of large change. Another area for improvement is the adaptive selection of the step size $\epsilon$ to optimally consume the distance and query budgets.

Given that our method has relatively few requirements, it is conceivable that it can be applied to any task for which the target model returns a continuous score for the prediction. For instance, speech recognition systems are trained to maximize the probability of the correct transcription, and neural network policies are trained to maximize some reward function over the set of actions conditioned on the current environment. A simple iterative algorithm that modifies the input at random elements may prove to be effective in these scenarios. We leave these directions for future work.

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

# S1 SUPPLEMENTARY MATERIAL

To demonstrate the generality of our evaluation of the Google Cloud Vision attack, we show 10 additional random images before and after perturbation by SimBA. In all cases, we successfully remove the top 3 original labels.

| origin_60.BMP | | | after_60.BMP | |
|---|---|---|---|---|
| Dog Breed | 93% | | Grass | 65% |
| Dog Like Mammal | 91% | | Snout | 63% |
| Dog | 90% | | Terrier | 60% |
| Lakeland Terrier | 85% | | | |
| Wire Hair Fox Terrier | 84% | | | |
| Terrier | 83% | | | |
| Snout | 62% | | | |
| Carnivoran | 62% | | | |
| Companion Dog | 60% | | | |

| origin_54.BMP | | | after_54.BMP | |
|---|---|---|---|---|
| Camera Accessory | 87% | | Weapon | 94% |
| Product | 82% | | Gun | 94% |
| Hardware | 67% | | Firearm | 76% |
| Optical Instrument | 66% | | Air Gun | 65% |
| Camera Lens | 61% | | Trigger | 63% |
| Gun | 61% | | Optical Instrument | 59% |
| Product | 58% | | Airsoft Gun | 58% |
| Weapon | 53% | | Rifle | 51% |

| origin_58.BMP | | | after_58.BMP | |
|---|---|---|---|---|
| Chicken | 98% | | Snow | 77% |
| Beak | 92% | | Winter | 69% |
| Galliformes | 91% | | | |
| Rooster | 89% | | | |
| Feather | 81% | | | |
| Fowl | 79% | | | |
| Poultry | 74% | | | |
| Bird | 70% | | | |
| Phasianidae | 58% | | | |

| origin_59.BMP | | | after_59.BMP | |
|---|---|---|---|---|
| Lighthouse | 98% | | Sky | 78% |
| Tower | 96% | | Sea | 73% |
| Landmark | 89% | | Computer Wallpaper | 56% |
| Beacon | 81% | | | |
| Sky | 78% | | | |
| Promontory | 77% | | | |
| Coast | 59% | | | |
| Sea | 58% | | | |
| Inlet | 56% | | | |

| origin_25.BMP | | | after_25.BMP | |
|---|---|---|---|---|
| Marine Mammal | 95% | | Fish | 94% |
| Fauna | 94% | | Ecosystem | 91% |
| Mammal | 93% | | Fish | 86% |
| Wildlife | 82% | | Shark | 75% |
| Terrestrial Animal | 80% | | Mouth | 69% |
| Organism | 77% | | Cartilaginous Fish | 67% |
| Snout | 74% | | Organism | 67% |
| Mouth | 71% | | Marine Biology | 63% |
| Whales Dolphins And Porpoises | 69% | | Jaw | 53% |

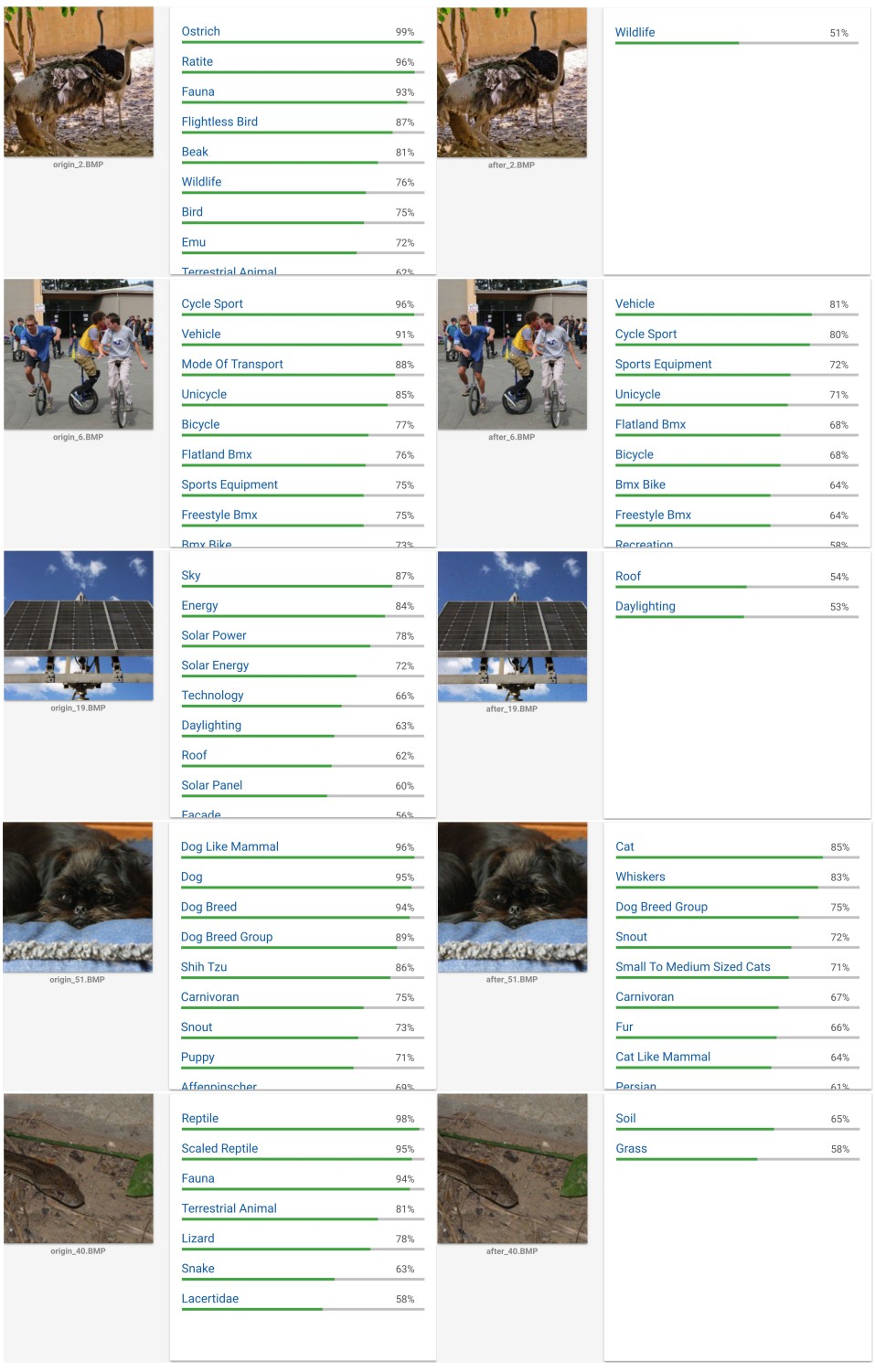

Figure S1: Additional adversarial images on Google Cloud Vision.

