# OpenReview forum: "Simple Black-box Adversarial Attacks"
_ICLR.cc/2019/Conference_

### Official Review · AnonReviewer2 · 2018-11-01
**interesting black-box adversarial attack using DCT basis; performance evaluation on targeted attack is insufficient and threat model is inconsistent (unfair comparison)**

**Rating:** 4
**Confidence:** 5

**Review:**

This paper proposed a simple query-efficient "score-based" black-box attack based on iteratively perturbing an input image with a direction randomly sampled (w/o replacement) from a set of orthonormal bases. In particular, the authors proposed the use of low-frequency parts of DCT (discrete cosine transformation) as in (Guo 2018) to perform this task. Experimental results on ImageNet and three different classification models demonstrate the query efficiency of the proposed method -- able to achieve high attack success rate within fewer query budgets, where the visual distortion has an L2 norm threshold set to be 10. The authors also demonstrate an untargeted score-based black-box attack on Google CloudVision API.

While the results seem promising, there are several issues that may potentially weaken the query-efficient claims made in this paper, especially due to the lack of sufficient attack comparisons (on smaller datasets) and inconsistent threat models when compared to existing works. My main concerns are summarized as follows.

1. Unfair comparison due to inconsistent threat models (knowledge known to an attacker): the proposed method (simBA) is a "score-based" black-box attack, not a "decision-based" black-box attack. The proposed method assumes knowing the prediction likelihood (or prediction score) as the model output when performing black-box attacks, whereas the compared methods in black-box settings, such as Opt Attack and Boundary Attack, are "decision-based" attack that assumes only knowing the top-1 prediction label. Therefore, the query count comparison is meaningless and unfair, since these two methods require far less information from the model.

On the other hand, ZOO/AutoZOOM is a score-based attack. But ZOO can achieve a very low L2 distortion due to its coordinate descent nature. A fair comparison is to set the same L2 distortion for all score-based methods, and compare the median/avg query counts of each image to reach the same L2 distortion. The comparison to Opt-Attack / Boundary attack makes sense only if the proposed method (simBA) can also perform decision-based attack. Nonetheless, the query count to same-distortion comparison argument still holds. The authors should specify whether simBA can apply to the decision-only attack scenario. If so, how to implement and what is the performance?

Lastly, the QL attack (Ilyas 2018) can perform both score-based and decision-based attacks. So the authors should make the query comparison (to same L2 distortion) as well. According to a recent report (Table-1, NES column) in https://arxiv.org/pdf/1807.07978.pdf, the QL-attack has a comparable performance in terms of query counts as reported in this paper.

2. More experiments on targeted black-box attacks: While untargeted attacks on Imagenet is a relatively easy task, I was a bit skeptical on the attacking performance of simBA in targeted attacks - since the selection of low-frequency bases directly limits the search space of adversarial examples, as opposed to arbitrary random directions adopted in QL-attack, Boundary-attack, and Opt-attack. It is also not clear how the target label is chosen in the targeted attack experiment.
I suggest including two more experiments to validate the function of simBA: (i) compare the performance of least-likely targeted attack (ii) show results on smaller datasets such as Cifar-10. As pointed out by the authors, Imagenet has too many image dimensions and make it more vulnerable to attack. Showing attacking results on smaller datasets can properly justify the value of the proposed attack, rather than the benefit from high dimensionality.


3. Novelty relative to LFBA (Guo) should be better differentiated: The idea of using DCT is originated from the LFBA paper. Since in that paper the authors also leveraged low-frequency DCT to perform black-box attacks, it is not clear to me what makes the proposed method perform better than the LFBA paper. The novelty and difference between this paper and the LFBA paper should be addressed.

4. The Google Cloud Vision API attack is not too appealing - the tree label is still there and the trees are obviously present in the picture, while I appreciate the effect of removing the original top-3 labels. Can the authors show another set of non-trivial (more surprising) and targeted-attack experiments? Or simply do the same experiment using the same image (men snowing -> dog)  as in the QL-attack.

----
Post-rebuttal review

I appreciate the authors' efforts in clarifying some of my concerns. However, I am still not convinced the comparison has been made fair. Many numbers from Table 1, such as ZOO, Opt-attack, QL-attack and AutoZOOM seem to be directly adapted from the papers rather than implemented and reproduced based on the same setting as the proposed attack. In particular, given that QL-attack is a published work, one of the state-of-the-art method and its codes has been released, I would really love to see a direct comparison using the same data samples and threat model. I would also like to emphasize that implementing all attacks under the same setting is crucial, since different attack methods may have a different criterion to determine attack successfulness. For example, QL-attack has some pre-defined distortion (L2 or Linfinity) for determining an adversarial example is successful, in addition to a different predicted class.

---

> ### Author Response · Authors · 2018-11-16
> **Re: interesting black-box adversarial attack using DCT basis; performance evaluation on targeted attack is insufficient and threat model is inconsistent (unfair comparison)**
>
> Our detailed response is below. We are happy to include the additional results and comparisons that you ask for, however we do want to emphasize that we think it is generally inappropriate to ask for a comparison with a concurrent submission to the same conference [Ilyas et al. 2018].
>
> Detailed response:
> We believe there may have been several possible misunderstandings that resulted in R2’s concerns.
>
> 1. We agree that the threat model varies across the various baselines. However, our inclusion of boundary attack and Opt attack are not meant to diminish their results, but rather to be comprehensive in our evaluation of prior work. As for other score-based attacks, although it is true that ZOO can sometimes achieve a very low L2 distortion, it also suffers from an order of magnitude higher failure rate (11.1% rather than <2% of SimBA) and requires several orders of magnitude more queries (192,000!!! rather than 1,232 for SimBA-DCT or 1,665 SimBA). In the preprint https://arxiv.org/pdf/1807.07978.pdf, the QL-attack is comparable to our attack in terms of query efficiency but at a high failure rate of 41.7%!! We believe that our evaluation is as fair to the other baselines as possible by achieving a lower L2 distortion than all methods other than ZOO and maintaining a success rate close to 100%, while requiring far fewer queries.
>
> 2. We performed targeted attack against random classes, similar to Tu et al. 2018 (https://arxiv.org/abs/1805.11770) and Cheng et al. 2018 (https://arxiv.org/pdf/1807.04457.pdf). We tested our attack against the least likely class as well and found that our method is less efficient but remains very competitive. More precisely, the average query count for SimBA increases from 7,899 to 12,256, while the average query count for SimBA-DCT increases from 9,275 to 17,272.
>
> We also tested our method on CIFAR-10 (targeted attack against the least likely class) and found that restricting to the low frequency basis does not affect the attack’s efficiency. SimBA achieves an average query count of 522 with average L2 norm = 1.41, while SimBA-DCT achieves an average query count of 606 with average L2 norm = 1.60. Both attacks are successful 100% of the time and are very competitive with state-of-the-art attack algorithms such as AutoZOOM. While using low frequency perturbations does not improve the attack for CIFAR-10, it does not hinder the attack’s efficiency either. As for the comment regarding limiting search space, Guo et al. have found that restricting to the low frequency subspace does not hinder adversarial optimality, which is empirically demonstrated in both theirs and our work.
>
> 3. The basic SimBA attack uses axis-aligned directions rather than the DCT basis when picking random directions of descent, and provides the majority of the query efficiency compared to other methods (see Table 1). The choice of the DCT basis further improves our attack in the untargeted case and demonstrates that it can generalize to other orthonormal bases, but is not crucial. In this regard, our paper differs significantly from the work of Guo et al.
>
> 4. Since our attack does not begin with an image of the adversarial class, it is not designed for targeted attacks on GCV. However, as GCV is the most widely used real world image classification platform for black-box adversarial attacks, we would like to demonstrate the efficacy of our method despite this limitation. Thus, we chose removing the top 3 original classes as a reasonable objective. In comparison with the QL-attack, the query efficiency of our method allows substantially more adversarial images to be created within the same cost budget, and our work is the first to show aggregate statistics for attacking a deployed machine learning service. We included 10 additional random samples in the Supplementary Material. For a non-trivial example, the second image in the Supplementary Material shows a case where a set of camera instruments is misclassified as a weapon after perturbation.

---

> > ### Comment · AnonReviewer2 · 2018-11-16
> > **Clarification on NES attack performance (aka QL attack)**
> >
> > I appreciate the authors' prompt response. But the authors apparently misunderstood my comment 1. I am NOT asking to compare with the new method proposed in an unpublished work (Ilyas 2018 - https://arxiv.org/pdf/1807.07978.pdf). Rather, I am asking the performance comparison with NES attack (it is called QL attack in this paper), which is a published paper at ICML 2018 (https://arxiv.org/pdf/1804.08598.pdf). I did explicitly point out the "NES column" in the table of the unpublished work because it's basically the same setting to be compared with the results presented in this paper. I am very surprised to see the response that "we think it is generally inappropriate to ask for a comparison with a concurrent submission to the same conference", as QL attack was already compared (but in a different setting) in this paper.

---

> > > ### Author Response · Authors · 2018-11-16
> > > **Re: Clarification on NES attack performance (aka QL attack)**
> > >
> > > We understand that the NES attack is the same as the QL-attack evaluated in our paper. We would like to point out that the evaluation in Ilyas et al. 2018 (https://arxiv.org/pdf/1807.07978.pdf) does not compromise our claim of unprecedented efficiency. Although QL-attack can achieve average query count close to that of our method, its failure rate is extremely high -- 41.7% -- whereas SimBA and SimBA-DCT achieve failure rate of <2% within around the same number of queries.

---

### Official Review · AnonReviewer3 · 2018-11-03
**simple and effective blackbox attack based on random directions**

**Rating:** 6
**Confidence:** 3

**Review:**

This paper presents a simple and effective black box adversarial attack on sota deep nets for image classification tasks. It is based on randomly picking a low frequency component of the DC Transform.  It is claimed to be most efficient when compared to the sota methods in terms of number of queries required for the attack. It is shown that a median of 600 queries for resnet-50 for imagenet dataset, and 2500 for google cloud vision. Due to its simplicity, it is also claimed that the attack is quite simple to implement in code. The paper presents a detailed analysis of their attack in pixel and DCT space, targeted vs untargeted attack, comparison over different architecture such as Densenet, resnet, and inception.

Though the work is quite important and presents a simple and effective baseline black box attack. My concern is primarily on the novelty and originality of the idea, as it is mainly based on the work of Guo etal 2018, which this paper says is the motivation behind their work. So, it is not clear what is the contribution of this paper, as a similar study seems to have been carried out in that paper as well. The authors do not clearly give the relative comparison wrt Guo etal 2018.

---

> ### Author Response · Authors · 2018-11-16
> **Re: simple and effective blackbox attack based on random directions**
>
> Comparison with Guo et al. 18:
> There may have been a misunderstanding. We do compare directly to the exact method by Guo et al. It is algorithm “LFBA” in Figure 2 and Table 1. LFBA stands for “Low Frequency Boundary Attack”, which is the terminology used by Guo et al.
>
> Novelty over Guo et al. 18:
> While black-box adversarial examples in DCT space has certainly been studied in Guo et al.,
> the core component that makes SimBA drastically more efficient than other black-box attacks is that we take numerous small steps in random orthonormal directions, whether that be random pixel perturbations or the DCT basis. The fact that SimBA (without DCT) is already very competitive compared to all previous untargeted and targeted black-box attacks supports this claim. We consider this insight important to be shared with the community, because it shows that the problem of adversarial attacks may be much simpler than most of us had assumed.

---

### Official Review · AnonReviewer1 · 2018-11-06
**simple algorithm, intriguing message**

**Rating:** 6
**Confidence:** 3

**Review:**

This paper demonstrates that a simple greedy random search algorithm in DCT space based on score feedback is able to synthesize adversarial examples with quite good query efficiency.  The algorithm is demonstrated on ImageNet  with three common architectures, showing much higher efficiency when sampling from the DCT basis. The algorithm is also shown to outperform state of the art attacks in terms of query count. Finally, a successful attack is demonstrated on Google Cloud Vision.

While not particularly heavy on technicalities, this work does make a couple intriguing points, namely that adversarial attacks can potentially be quite easy to perform due to the inherent nature of high-dimensional classification, and that the space is in which the search is perform might be more important than the sophistication of the search itself. I interpret the proposal not so much as a claim to a state-of-the-art algorithm (even though the results are impressive) but as a very reasonable baseline in the evaluation of attack efficiency -- one might even wonder why it has not been common practice thus far to evaluate against such kinds of algorithms by default.

---

> ### Author Response · Authors · 2018-11-16
> **Re: simple algorithm, intriguing message**
>
> Thank you for your review. We are equally surprised that the SimBA attack has not been discovered earlier - and that it can even outperform far more sophisticated approaches.

---

### Author Response · Authors · 2018-11-16
**Revision**

We have uploaded a revision with the following changes:

- Significantly improved attack when using the standard basis (SimBA). In particular, Table 1 and Figures 2-4 were updated.
- Supplementary material containing 10 additional sample images for Google Cloud Vision attack.

---

### Meta-Review · Area_Chair1 · 2018-12-16
**A method that is appealing for its simplicity, but reviewer concerns regarding fairness of comparison persist.**

**Confidence:** 3
**Recommendation:** Reject

**Metareview:**

The paper considers a procedure for the generation of adversarial examples under a black box setting. The authors claim simplicity as one of the main selling points, with which reviewers agreed, while also noting that the results were impressive or "promising". There were concerns over novelty and some confusion over the contribution compared to Guo et al, which I believe has been clarified.

The highest confidence reviewer (AnonReviewer2), a researcher with significant expertise in adversarial examples, raised issues of inconsistent threat models (and therefore unfair comparisons regarding query efficiency), missing baselines. A misunderstanding about comparison against a concurrent submission to ICLR 2019 was resolved on the basis that the relevant results are mentioned but not originally presented in the concurrent submission.

While I disagree with AnonReviewer2 that results on attacking a particular image from previous work (when run against the Google Cloud Vision API) would be informative, the reviewer has remaining unaddressed concerns about the fairness of comparison (comparing against results reported in previous work rather than re-run in the same setting), and rightly points out that as many variables should be controlled for as possible when making comparisons. Running all methods under the same experimental setting with the same *collection* of query images is therefore appropriate.

The authors have not responded to AnonReviewer2's updated post-rebuttal review, and with the remaining sticking point of fairness of comparison with respect to query efficiency I must recommend rejection at this point in time, while noting that all reviewers considered the method promising; I thus would expect to see the method successfully published in the near future once issues of the experimental protocol have been solidified.